# A Novel *KCNN2* Variant in a Family with Essential Tremor Plus: Clinical Characteristics and In Silico Analysis

**DOI:** 10.3390/genes14071380

**Published:** 2023-06-29

**Authors:** Maria d’Apolito, Caterina Ceccarini, Rosa Savino, Iolanda Adipietro, Ighli di Bari, Rosa Santacroce, Maria Curcetti, Giovanna D’Andrea, Anna-Irma Croce, Carla Cesarano, Anna Nunzia Polito, Maurizio Margaglione

**Affiliations:** 1Medical Genetics, Department of Clinical and Experimental Medicine, University of Foggia, 70122 Foggia, Italy; maria.dapolito@unifg.it (M.d.); cceccarini@ospedaliriunitifoggia.it (C.C.); iadipietro@ospedaliriunitifoggia.it (I.A.); ighli.dibari@unifg.it (I.d.B.); rosa.santacroce@unifg.it (R.S.); maria.curcetti@unifg.it (M.C.); giovanna.dandrea@unifg.it (G.D.); icroce@ospedaliriunitifoggia.it (A.-I.C.); ccesarano@ospedaliriunitifoggia.it (C.C.); 2Neuropsychiatry for Child and Adolescent Unit, Department of Woman and Child, Policlinico Riuniti, 70122 Foggia, Italy; rsavino@ospedaliriunitifoggia.it (R.S.); apolito@ospedaliriunitifoggia.it (A.N.P.)

**Keywords:** gene, *KCNN2*, SK2 channel, developmental delay, essential tremor

## Abstract

Background: Essential tremor (ET) is one of the more common movement disorders. Current diagnosis is solely based on clinical findings. ET appears to be inherited in an autosomal dominant pattern. Several loci on specific chromosomes have been studied by linkage analysis, but the causes of essential tremor are still unknown in many patients. Genetic studies described the association of several genes with familial ET. However, they were found only in distinct families, suggesting that some can be private pathogenic variants. Aim of the Study: to characterize the phenotype of an Italian family with ET and identify the genetic variant associated. Methods: Clinical and genetic examinations were performed. Genetic testing was done with whole-exome sequencing (WES) using the Illumina platform. Bidirectional capillary Sanger sequencing was used to investigate the presence of variant in all affected members of the family. In silico prediction of pathogenicity was used to study the effect of gene variants on protein structure. Results: The proband was a 15-year-old boy. The patient was the first of two children of a non-consanguineous couple. Family history was remarkable for tremor in the mother line. His mother suffered from bilateral upper extremity kinetic tremors (since she was 20 years old), anxiety, and depression. Other relatives referred bilateral upper extremity tremors. In the index case, WES analysis performed supposing a dominant mode of inheritance, identified a novel heterozygous missense variant in potassium calcium-activated channel subfamily N member 2 (*KCNN2*) (NM_021614.3: c.1145G>A, p.Gly382Asp). In the pedigree investigation, all carriers of the gene variant had ET and showed variable expressivity, the elder symptomatic relative showing cognitive impairment and hallucinations in the last decade, in addition to tremor since a young age. The amino acid residue #382 is located in a transmembrane region and in silico analysis suggested a causative role for the variant. Modelling of the mutant protein structure showed that the variant causes a clash in the protein structure. Therefore, the variant could cause a conformational change that alters the ability of the protein in the modulation of ion channels Conclusions: The *KCNN2* gene variant identified could be associated with ET. The variant could modify a voltage-independent potassium channel activated by intracellular calcium.

## 1. Introduction

Tremor is defined as an involuntary and rhythmic oscillatory movement of a body part, and may be categorised as rest, postural, or kinetic, depending on when it manifests maximally. Common causes include essential tremor (ET), Parkinson’s disease, dystonia, metabolic disorders, drug-induced tremor, and enhanced physiological tremor [1,2,3].

ET is considered as one of the more prevalent movement disorders [4]. Although ET is a distinct clinical entity and it is often misdiagnosed, particularly as Parkinson’s disease. ET main clinical manifestations involve the upper limbs (approximately 95% of patients), and less commonly the head, face, voice trunk, and lower limbs. Current diagnosis is merely based on clinical findings, which poorly differentiate ET from similar phenocopies and other conditions associated with tremors. The pathophysiological causes of essential tremor are largely unknown. A number of studies suggested that ET is caused by a functional disorder of the oscillatory network comprising the cerebello–thalamo–cortical pathway [5]. Findings from studies using animal models, human tissues, and brain imaging suggested a pivotal role for the gamma-aminobutyric acid (GABA) in the pathogenesis of ET [6]. However, there are several arguments against the GABA hypothesis in ET [7].

Harmaline-induced tremor in rodents is a useful animal model to study ET [8]. Findings on this model suggest that the mechanism of tremor is the modulation of the T-type Ca2^+^ channel. In addition, the selective knockdown of the *Cav3.1* gene expression in the inferior olive neurons efficiently suppressed harmaline-induced tremor in wild-type mice [9].

About fifty percent of the ET cases are thought to be caused by a genetic risk factor. Familial forms of ET often appear early in life. Several loci on different chromosomes that may be linked to essential tremor have been studied, but no specific genetic associations have been confirmed. Several genes and environmental factors seem to be associated with an individual risk of developing this complex condition.

ET appears to be inherited in an autosomal dominant pattern, in which one copy of a mutated gene is sufficient to cause the disorder [10]. Technological advances in genomics allowed for progress in understanding the genetic etiology of many diseases and also contributed to remarkable advances in our understanding of ET pathological mechanisms. Several loci on specific chromosomes have been studied and at least six disease loci of ET identified: (ETM1 (3q13, OMIM 190300), ETM2 (2p24.1, OMIM 602134), ETM3 (6p23 OMIM 611456), ETM4 (16p11.2 *FUS* gene OMIM 614782), ETM5 (11q14.1 *TENM4* gene OMIM 610084), and ETM6 (1q21.2 *NOTCH2NLC* gene OMIM 618866). Linkage analysis identified physical segments of the genome that cosegregate with ET phenotypes through families. This type of study identified four susceptibility loci for familial ET, which have been located at chromosome 2p25-p22 [10], 3q13 [11], 5q35 [12], and 6p23 [13]. Genome-wide association studies (GWAS) described several variants in *LINGO1*, *SLC1A2*, *STK32B*, *PPARGC1A*, and *CTNNA3* but none of them have been confirmed in replication studies [14,15,16]. Exome studies showed the association of several genes with familial ET (*FUS* (currently designated as ETM4) [17], *HTRA2* [18], *TENM4* (designated as ETM5) [19], *SORT1* [20], *SCN11A* [21], *NOTCH2NLC* (designated as ETM6) [22], *CACNA1G* [23], *NOS3*, *KCNS2, HAPLN4, USP46, SLIT3, CCDC183*, *MMP10*, and *GPR151* [24,25,26]. However, they were found only in a single family suggesting that some pathogenetic variants can be private. In this report, we present clinical and genetic characterization of an Italian family presenting with ET. Genetic testing was performed with whole-exome sequencing (WES) using Illumina platform. The effect of the identified gene variant was investigated by using in silico prediction of pathogenicity and within family linkage analysis.

## 2. Materials and Methods 

### 2.1. Clinical Investigation

We performed clinical and genetic investigations on a 15-year-old boy who was referred to Neuropsychiatric Unit for Child and Adolescent at the Policlinico Riuniti Hospital of Foggia, due to the occurrence of bilateral upper extremity tremors. Physical and neurological examination, wake and sleep electroencephalography (EEG), cardiac, abdominal, and neck ultrasound, motor and sensory electroneurography of the bilateral ulnaris and medianus nerves, and brain magnetic resonance imaging (MRI), were performed according to a consensus of the Task Force on Tremor of the International Parkinson and Movement Disorder Society, 2018 (IPMDS) [27]. All family members available for assessment underwent a detailed medical neurological examination. Clinical and genetic studies were conducted in agreement with the Helsinki declaration. All patients investigated received genetic counselling and provided informed consent for testing. All data presented in the manuscript were accurately anonymized. The study was approved by the local Ethics Committee. (Protocol code 17/CE/2014).

### 2.2. FMR1 Analysis

Screening for FMR1 expanded alleles was performed using a capillary electrophoresis technique (FraxA 1 Kit, Experteam srl, Venice, Italy).

### 2.3. Exome Investigation

WES investigation was performed only on proband because family history investigation suggested a dominant inheritance. Indeed, according to our internal rules, when a dominant inheritance is assumed, a trio WES, including proband’s parents, is not perform. Genomic DNA isolation from the whole blood of proband was conducted in according to standard methods [28]. The Illumina DNA Prep with Enrichment protocol (San Diego, CA, USA) was used to prepare sample libraries from peripheral blood-purified genomic DNA. All coding regions and exon–intron junctions (±50 bps) were sequenced using an Illumina NextSeq 550^®^ Sequencing System (San Diego, CA, USA). We obtained about 99% of targeted regions covered more than 30 times and 99 × mean depth. The data set generated by the NextSeq 550 System were then processed according to the Genome Analysis Toolkit (GATK 1.6), annotated, and were analysed using the software BaseSpace Variant Interpreter Annotation Engine 3.15.0.0 (Illumina, San Diego, CA, USA). Variants annotations were based on the Human Genome Variation Society guidelines (HGVS), mapped to the human genome build GRCh37/UCSC hg19, and classified allowing the criteria of the American College of Medical Genetics and Genomics [29]. Classification of pathogenicity for all rare genetic variants was performed according to ACMG2015 [29]. Variant prioritization was carried out using, as an initial filter, minor allele frequency (MAF) in the Exome Aggregation Consortium (ExAC) 0.01. Additional investigation was done to identify variants pathogenic, likely pathogenic, or VUS associated with a given phenotype.

### 2.4. Sanger Sequencing

According to recommendation of the American College of Medical Genetics and Genomics, Sanger resequencing on SeqStudio™ Genetic Analyzer System Applied Biosystems™ (Waltham, MA, USA) was used to confirm the WES results. The *KCNN2* gene (NM_021614.4) exon 5 was amplified using specific primers and PCR products were then sequenced using BigDye Terminator v.3.1 (Thermo Fisher Scientific, Waltham, MA, USA), according to standard protocols [30].

### 2.5. In Silico Analysis and Protein Structure Modeling

To further prioritize the candidate gene variants, nonsynonymous amino acid substitutions and splicing variants were selected on the basis of annotations on protein functionality and information of the phenotype. Different available bioinformatic tools were used to investigate pathogenicity of genetic variants on protein functionality. Effects of the p.Gly382Asp substitution were explored by using in silico pathogenicity prediction tools (PolyPhen-2, http://genetics.bwh.harvard.edu/pph2/ (accessed on 21 March 2023); SIFT, http://sift.jcvi.org/siftbin/retrieve_enst.pl (accessed on 18 November 2022); and mutation Assessor (http://mutationassessor.org/r3) (accessed on 21 March 2023). REVEL and MetaLR pathogenicity scores are taken from dbNSFP (version 4.3a, https://sites.google.com/site/jpopgen/dbNSFP, accessed on 21 March 2023), a database for functional prediction and annotation of all potential nonsynonymous single nucleotide variants. The CADD (combined annotation-dependent depletion) tool (http://cadd.gs.washington.edu/home (accessed on 21 March 2023) was used to integrate multiple annotations for the deleteriousness of single nucleotide variants, insertion/deletion variants in the human genome. The KCNN2 AlphaFold structure prediction model (AF-Q9H2S1-F1-model_v4.pdb) was then used as a template to examine the putative pathogenic effect of the substitution by using MISSENSE3D (http://missense3d.bc.ic.ac.uk), DynaMut (https://biosig.lab.uq.edu.au/), and Swiss-Pdb Viewer. 

## 3. Results

### 3.1. Proband Clinical Characteristics

The index case was a 15-year-old boy referred to Neuropsychiatric Unit for Child and Adolescent due to the occurrence of bilateral upper extremity tremors shaking that had started insidiously and progressively worsened over the past year. He described this tremor as a prevalent kinetic tremor, which increases during different voluntary movements (such as writing, eating, etc.) and in the case of stress or anxiety. The patient was the first of two children of a non-consanguineous couple. At the time of conception, his mother was twenty-seven and his father was thirty. His sister was ten years old. The proband was born at term through a caesarian section for sudden placental abruption. The Apgar score was in the normal range. No signs of perinatal distress were reported. Breastfeeding occurred up to 8 months of age. No delay in motor and language development was reported. Daytime and nighttime sphincter control was acquired at 4 years of age. Sleep and feeding have always been regular. As well, his social skills and learning abilities were described to be in line with his peers. Auxological evaluation was in the normal population range (height 76 kg, weight 170 cm). Physical examination did not reveal any evident dysmorphic features. Neurologic examination showed slight clumsiness, and bilateral upper limbs tremors, which were more evident during Mingazzini I and Finger-to-nose tests. Muscle size, tone, and strength were normal. Hyperactive deep tendon reflexes were detected bilaterally at the upper and lower limbs. A complete blood count and the comprehensive metabolic panel did not show any abnormalities with the exception of a mild increase of insulin plasma levels (49.2 mcUl/mL; n.v. 3.2–16.3). Electrolytes, kidney, liver, and thyroid function tests were within the normal range. Wake and sleep electroencephalography (EEG), cardiac, abdominal, and neck ultrasound were normal. Motor and sensory electroneurography of the bilateral ulnar and median nerves revealed physiological nerve conduction. Brain magnetic resonance imaging (MRI) showed non-specific small gliotic-like foci in the parietal subcortical and small periventricular ectasias mainly in the subnuclear and left semioval center. Furthermore, a slight descent of the right cerebellar tonsil was detected, without any pathological significance. Spectrometry of the basal nuclei did not display altered ratios between metabolites. Neuropsychological evaluation was also conducted. The evaluation of cognitive functioning and intellectual abilities revealed a composite Q.I. score of 66 (Wechsler Intelligence Scale for Children 4° edition: WISC-IV). It provided the following primary subtest scores: Verbal Comprehension Index (VCI), 78; Visual–Spatial index (VSI), 58; Working Memory Index (WMI), 88; and Processing Speed Index (PSI), 79. The General Ability Index (GAI) was 64, while Cognitive Proficiency Index (CPI) scored 79. To verify and quantify the severity of depression and anxiety symptoms, the patient was administered Children’s Depression Inventory-Self Report, Second Edition (CDI-2), and Multidimensional Anxiety Scale for Children-Self Report, Second Edition (MASC-2), which showed higher levels of anxiety symptoms (MASC T-score = 60), mainly for Separation Anxiety/Phobias, Social Anxiety (Performance Fears), and Obsession and Compulsions. According to the Consensus Statement on the Classification of Tremors criteria (from the Task Force on Tremor of the International Parkinson and Movement Disorder Society, 2018), [27] clinical and neuropsychological assessments led to the diagnosis of “Indeterminate tremor Syndrome”, because the patient showed a familiar, action tremor (both postural and intention tremor), of less than 3 years of duration, associated with a mild cognitive impairment (mainly in the visual–spatial domain) and anxiety.

### 3.2. Pedigree Investigation

Family history was remarkable for tremor in the mother line. We studied the Italian family with four symptomatic individuals over three generations (Figure 1). Symptoms and clinical signs in all affected family members are presented in Table 1. 

The proband’s mother (IV-3) suffered from essential tremor, mainly involving her upper extremities, anxiety, and depression. She was 43-year-old at the time of the examination. The evaluation of cognitive functioning and intellectual abilities revealed a composite Q.I. score of 77, resulting in borderline intellectual functioning (WISC-IV). Primary subtest scores provided were: Verbal Comprehension Index (VCI), 80; Perceptual Reasoning Index (PRI), 75; Working Memory Index (WMI), 77; and Processing Speed Index (PSI), 97. Even the mother’s cognitive profile highlighted a worse impairment in the visual–spatial domain. This mild cognitive impairment associated with essential tremor met the criteria for the “Essential tremor plus” diagnosis. Clinical investigation of mother’s cousin (IV-5) reported upper extremity tremors and anxiety. The maternal aunt (III-3) suffered from upper extremity tremors and anxiety since she was 20 years old. Indeed, she presented a bilateral akinetic-rigid syndrome characterized by loss of motor function, more evident in the left limbs, and tremor at rest. The patient also had severe cognitive impairment and visual hallucinations (Lewy body dementia (LBD)). Over the past two years, she needed assistance in the normal activities of daily life. In addition, unspecified bilateral upper extremity tremors were reported in maternal great-grandfather (II-6) and grandfather (III-2). 

### 3.3. Genetic Studies

As up to 2% of patients with suspected ET and additional neurological features suffer from Fragile X–associated tremor/ataxia syndrome (FXTAS) [31], FMR1 CGG repeat expansion mutation detection analysis was performed in the proband and his mother. In both of them, genetic analysis revealed the presence of 30 (±1) repeats and excluded the presence of FMR1 premutation in the proband and his mother. 

WES of the index case, performed assuming a dominant mode of inheritance, generated a large number of variants (12,114), of which 11,565 single nucleotide variations and 631 indels were found. Variants were classified (i.e., exonic: intronic, and untranslated regions; exonic: synonymous, nonsynonymous, stop gain/loss, frameshift, allele frequency; and so on) and prioritized using open-source software (Variant Interpreter, Illumina, San Diego, CA, USA).

To reduce the number of potentially deleterious gene mutation, we applied the follow filtration steps: (1) The initial filter considered sequencing quality controls with a quality score of greater than 30; (2) exclusion variants with a minor allele frequency of greater than 0.01 after querying the single nucleotide polymorphism database (http://www.ncbi.mln.nih.gov/snp (accessed on 21 March 2023), Exome Aggregation Consortium (http://exac.broadinstitute.org/ (accessed on 21 March 2023), Exome Variant Server (http://evs.gs.washington.edu/EVS (accessed on 18 November 2022), 1000 Genomes Projects (http://browser.1000genomes.org (accessed on 21 March 2023), and previous publication; (3) removing variants in intronic regions or synonymous coding variants; (4) selecting variants that segregate according to the supposed pattern of inheritance; and (5) querying disease databases, such as ClinVar (http://www.ncbi.nlm.nih.gov/clinvar (accessed on 18 November 2022), OMIM, (http://www.omim.org (accessed on 21 March 2023), and the Human Gene Mutation Database locus-specific database (http://www.hgvs.org/ (accessed on 21 March 2023) to further prioritize the candidate gene variants. Many of these variants were removed by looking at function of the related protein and the associated phenotype reported in literature. After the functional annotation step, approximately 315 variants were retained. 

Finally, the analysis of variants generated within BaseSpace Variant Interpreter was restricted to virtual subpanels using a gene list associated to the disease (Appendix A). After this step, only 24 variants remained. Of them, 23 resulted benign or likely benign according to ClinVar annotation, and only a heterozygous variant in the *KCNN2* gene remained to be further investigated. No variants involving other known tremor susceptibility genes were detected (see Appendix A).

*KCNN2* encodes the well-described 579 amino acid SK2 channel subunit (Q9H2S1-1 in Uniprot). The NCBI GenBank accession number NM_021614.3 (Ensembl: ENST00000264773) genetic sequence was used as reference sequence. A very low tolerance for loss of function variants of the gene was reported (pLoF = 0.99). The variation found was a nucleotide transition (c.1145G>A) and predicted a missense amino acid change occurring in the *KCNN2* exon 5, the substitution of a glycine by an aspartic acid (p.G382D). The gene variant (rs757982546) was not reported in ClinVar database. The variant identified was rare, being reported in GnomAD_exome and ExAC with an allele frequency of 0.000004 and 0.000001, respectively. PolyPhen-2, SIFT, REVEL, MetaL, Mutation Assessor, and combined annotation-dependent depletion (CADD) scores were calculated to evaluate the pathogenicity of the identified variation (Table 2). Results from in silico analyses showed that the variant could be deleterious. A detrimental effect of the variant was also predicted by the high CADD score of 32. 

According to standard protocol, the variant p.G382D identified using WES was confirmed in the proband and investigated in six family members by direct capillary Sanger sequencing (Figure 1B). After Sanger sequencing confirmation and genotyping of all available family members, the *KCNN2* variant G382D resulted to completely segregate with the phenotype (Figure 1B). The comparison of the human SK2 protein with ortholog sequences revealed a high conservation among different species (Figure 1C). The amino acid residue #382 is located in the functional SK2 transmembrane domain S6 (Figure 1D). 

The KCNN2 AlphaFold structure prediction model (AF-Q9H2S1-F1-model_v4.pdb) was used as a template to investigate the putative pathogenic effect of the p.G382D variant. The in silico analysis (MISSENSE3D; http://www.sbg.bio.ic.ac.uk/missense3d (accessed on 18 November 2022) indicated structural damage. The substitution replaced a buried uncharged residue (GLY, RSA 0.0%) with a charged residue (ASP, RSA 1.2%). Swiss-Pdb Viewer modelling shows comparison of the predicted structures of both wild-type and mutant proteins (Figure 2) (https://spdbv.unil.ch/) (accessed on 21 March 2023). Swiss-Pdb Viewer detected protein structural clashes due to aspartic acid substitution. In order to evaluate the effect of the identified variant on the residual flexibility, we performed in silico analysis using DynaMut web server. This tool predicted a vibrational entropy energy change between the wild-type (G382) and the variant (D382) protein, resulting in a decrease of molecule flexibility (ΔΔSVib ENCoM: −0.815 kcal·mol^−1^·K^−1^).

## 4. Discussion

The main purpose of this study was to investigate the pathogenesis of an autosomal dominant essential tremor in an Italian pedigree with multiple affected individuals. A novel heterozygous missense variant in the potassium calcium-activated channel subfamily N member 2 (*KCNN2*) gene was identified. 

ET is the most common movement disorder, but the pathogenic causes of definitive cellular pathways linked to ET still remain unclear. Several studies recognize ET as a hereditary disease with autosomal transmission. Clinical manifestations and age at onset can vary suggesting different penetrance and expression. A series of ET loci and candidate genes has been identified but, in most cases, pathogenic mechanisms remain to be understood. Indeed, known causative variants only explain a small portion of cases, suggesting the possibility of phenocopies, also within a family. 

In the present study, we used a WES approach to study the ET proband and his relatives. We demonstrated that all symptomatic members were heterozygote for the *KCNN2* G382D variant. No variant in additional known tremor susceptibility genes was detected. The variant identified was a nucleotide substitution c.1145G>A predicting a missense amino acid change, which implicated the substitution of a glycine by an aspartic acid, p.G382D. In silico analysis predicted a deleterious effect of the variant on protein structure and functionality. *KCNN2* encodes for the SK2 ion channel, a voltage-independent potassium channel activated by intracellular calcium [32,33,34], widely expressed in the central nervous system. This gene is of particular interest because it regulates neuronal excitability and synaptic transmission [34]. Calcium (Ca2^+^)-activated potassium channels (KCa) are a group of 6/7-TM ion channels that selectively transport K^+^ ions across biological membranes. They are activated by intracellular Ca2^+^, resulting in potassium efflux. SK2 channels play a critical role in synaptic transmission and neuron excitability by repolarizing or hyperpolarizing the membrane. Like other members of the SK channel family, the SK2 ion channel contains six hydrophobic transmembrane domains (S1–S6) and assembles to form homo- or hetero-tetrameric channel. The amino acid residue #382 is located in the functional SK2 transmembrane domain S6. The *KCNN2* variant G382D was predicted to cause a structural change in the transmembrane domain S6 that could disrupt normal protein transmembrane localization and affect depolarization and hyperpolarization of the membrane. SK channels are activated via calmodulin, which is constitutively linked to a C-terminal region of each subunit (calmodulin binding domain, CaMBD), and triggers the channel to open upon Ca2^+^ binding [35,36].

Variants in the *KCNN2* gene have been recently associated with neurological diseases. A heterozygous missense variation was found in a family with autosomal dominant myoclonus-dystonia (DYT34 OMIM 619724). The core phenotype consisted of childhood-onset dystonia that predominantly affected hands and neck, with a fast tremor and superimposed myoclonus. Furthermore, subtle cerebellar signs were reported in some patients. Whole-exome sequencing led to the identification of a heterozygous variant (c.1112G>A: p.Gly371Glu), located between the S5 and S6 domains in the pore region. The variant was shown to segregate with myoclonus-dystonia and anxiety in the three-generation family investigated [37]. Using exome sequencing, a de novo *KCNN2* frameshift deletion was identified in a patient with learning disabilities, cerebellar ataxia, and white matter abnormalities on brain MRI. In the same study, additional patients with de novo *KCNN2* variants leading to a SK2 channel loss-of-function were reported. All patients with pathogenic *KCNN2* variants had motor and language developmental delay, intellectual disability often associated with early-onset movement disorders encompassing cerebellar ataxia, and/or extrapyramidal symptoms (NEDMAB OMIM 619725) [38].

*KCNN2* has been considered a candidate gene for ET because it regulates neuronal excitability, synaptic transmission modulating intracellular calcium, and depolarization/hyperpolarization of the membrane [35]. Recently, a N-ethyl-N-nitrosourea (ENU)-induced mutant rat was described as a model for ET [39]. The tremor dominant Kyoto rat (F344-Trdk/+) is characterized by autosomal dominant whole-body tremor, especially evident around 2 weeks old and persistent throughout life. Tremor duration and intensity were significantly reduced with the administration of drugs used for ET treatment, but not with drugs used to treat Parkinson’s disease-related tremor. This indicates that the pharmacological phenotype of F344-Trdk/+ rats was similar to human ET. Positional candidate approach, genetic linkage analysis, and subsequent gene sequencing led to the identification of a heterozygote mutation (c.866T>A, p.I289N) in *Kcnn2* as the causative gene variant of the F344-Trdk/+ phenotype. In vitro electrophysiological studies demonstrated that the I289N mutation diminished SK2 channel activity. These findings suggested that *Kcnn2*, an ortholog of human *KCNN2*, is the causative gene for the tremor phenotype in F344-Trdk/+ rats. To our knowledge, associations of *KCNN2* variants with the risk for ET have not been found to date in humans. Using in silico prediction of pathogenicity and within family linkage analysis, the present study is the first in which a *KCNN2* gene variant was associated with ET. 

The high variability of clinical phenotypes associated with *KCNN2* gene pathogenic variants can be ascribed to a range of factors, including common variants, as well as in different genes, variants in regulatory regions, epigenetics, environmental, and lifestyle factors. For almost all human genetic diseases, individual variability in the phenotype is influenced by background variation in the genome. Variability can be due to the location and type of gene variant, with missense and protein-truncating variants in the same gene often causing different phenotypes.

It is conceivable that, while de novo *KCNN2* variants found in infancy-onset development delay patients with NEDMAB could have a high disruptive effect, the variant that we identified in the ET family investigated in the present study may have a milder effect on the normal protein function and cause a milder clinical phenotype. In keeping with this, in a cohort of patients investigated for *KCNN2* gene variants, a patient (#8) carrying a variation located in the S6 segment (p.Leu388Val) showed a mild clinical phenotype [38].

## 5. Limitations

The present study has some limitations because functional data, e.g., patch-clamp experiments, are lacking. However, genotype-phenotype segregation within the family, very rare frequency of the variation (GnomAD_exome: 0.000004 and ExAC: 0.000001), very low tolerance for loss of function variants (pLoF = 0.99) of the *KCNN2* gene, in silico prediction of a deleterious effect, and findings from animal rat model overlapping with human clinical phenotype support the causative role of the *KCNN2* p.G382D heterozygous variant. The present study evaluated only an informative family and results cannot be generalized. 

## 6. Conclusions 

We conclude that the rare *KCNN2* variant c.1145G>A, p.G382D, likely contributes to ET pathology. A series of clinical and laboratory findings provides initial evidence, but not definitive proof, that the *KCNN2* p.G382D variant plays a major effect in giving rise to an ET phenotype. This variant could modify the formation of multimeric protein complexes, affecting ion channels in the central nervous system utilized.

## Figures and Tables

**Figure 1 genes-14-01380-f001:**
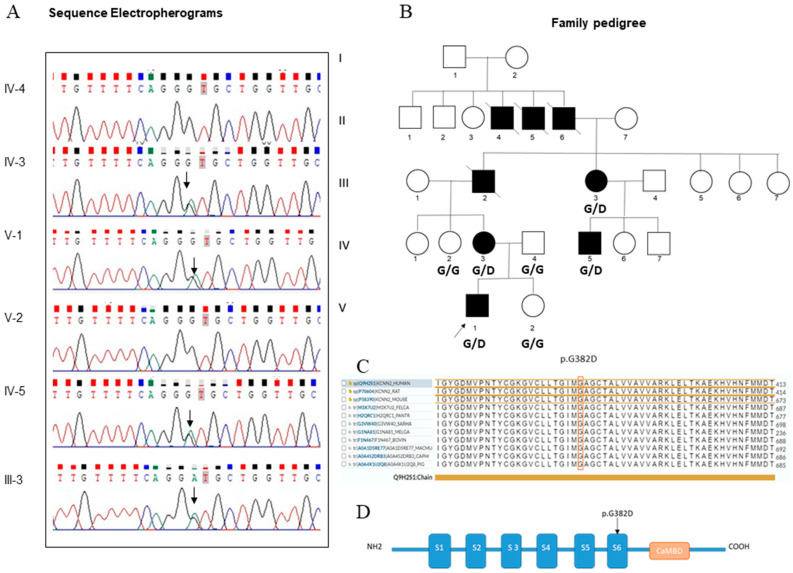
(**A**) Sanger sequencing showing the c.1145G>A substitution in the index case and family members. The arrow indicates the nucleotide change. (**B**) Pedigree of Italian family carrying the *KCNN2* variant. Circles show females, squares show males. Open symbols represent asymptomatic individuals. The genotype of investigated relatives is indicated: G/G: G382 homozygote; G/D: G382D heterozygote. (**C**) Alignment of the S6 domain (the amino acid position #382 is highlighted in red) along different species. (**D**) Schematic representation of the SK2 channel showing the localization of the pathogenic variant found in the proband and his family members.

**Figure 2 genes-14-01380-f002:**
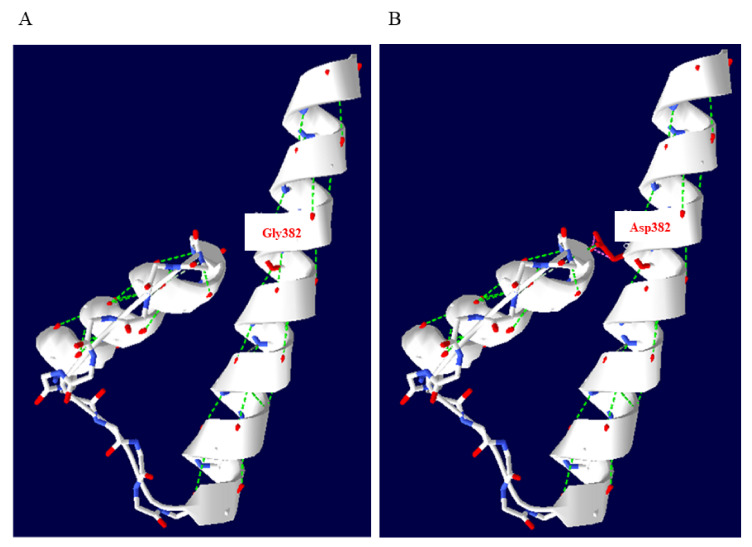
Comparison of the predicted structures of both (**A**) wild-type (G382) and (**B**) variant protein (D382) using Swiss-Pdb Viewer (https://spdbv.unil.ch/ accessed on 21 March 2023). Clashes are displayed in pink dotted lines in the mutant protein.

**Table 1 genes-14-01380-t001:** Summary of the clinical characteristics of affected family members.

	V-1 (Index Case)	IV-3	IV-5	III-3
Sex, age (years)	M, 15	F, 42	M, 40	F, 70
Age at onset (years), First symptom	14 upper extremity tremors—anxiety	20 upper extremity tremors—anxiety	15 upper extremity tremors—anxiety	20 upper extremity tremors—anxiety
X Examination	-Upper extremity tremors -Worse impairment in the visual–spatial domain -Anxiety	-Upper extremity tremors -Worse impairment in the visual–spatial domain -Anxiety -Depression	-Upper extremity tremors -Anxiety	-Upper extremity tremors -Lewy body dementia (LBD)

**Table 2 genes-14-01380-t002:** In silico predicted pathogenicity of the KCNN2 variant.

Gene	Nucleotide	Amino Acid Change	SIFT	PoliPhen	CADD	REVEL	MetaLR	Mutation Assessor
*KCNN2*	GGT/GAT	G382D	0	0.969	32	0.709	0.58	0.948

## Data Availability

The data presented in this study are available on request from the corresponding author.

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
