# Peer review of "A Novel *KCNN2* Variant in a Family with Essential Tremor Plus: Clinical Characteristics and In Silico Analysis"

_genes, 2023, doi:10.3390/genes14071380_

Round 1
Reviewer 1 Report
This is an interesting report of what appears to be a rare novel KCNN2 variant in a family with “essential tremor”. The authors did a good job of classifying the affected family members according to the consensus criteria of the International Parkinson and Movement Disorder Society.
The paper needs some minor editing to correct English grammar errors.
In the Abstract, I do not understand the sentence “Modelling of the resulting protein structure predicted that the variant disrupts structure and cause clash a likelihood of being pathogenic.” Please revise.
In the Introduction, the following is inconsistent with the current syndromic definition of essential tremor. I would delete all of it. “In addition to tremor, patients with ET showed additional clinical feature. These include bradykinesia or non-motor symptoms, depression, anxiety, cognitive impairment, fatigue, personality changes, olfactory dysfunction, hearing impairment, temperature extremes, sleep disturbances, and upper airway dysfunction.”
Line 54, page 2: “half percent” should be “fifty percent”???
In Table 1: I am not sure what is meant by “worse impairment in the visual-spatial domain”. Is this “visually guided kinetic tremor” or does this refer to cognitive function? Was cognitive impairment linked with the genetic variant.
All of the affected family members except the proband had ET plus, and the proband had indeterminant tremor due to the short history of tremor (1 year). The proband would eventually be classified as ET plus if his exam remained unchanged for an additional two years. Subject III-3 had antecedent ET or more likely ET plus. Did III-3 have a history of low intelligence? “Essential Tremor” in the title probablu should be changed to “Essential Tremor Plus”.
There is no indication of how severe the tremor was in any of the affected family members. Were clinical ratings performed?
See comments to authors.
Author Response
We thanks the Reviewer for her/his comments that helped us to ameliorate the presentation of results. We made further changes, and we are confident that the present version of the manuscript is strongly improved.
Major points
Point 1: In the Abstract, I do not understand the sentence “Modelling of the resulting protein structure predicted that the variant disrupts structure and cause clash a likelihood of being pathogenic.” Please revise.
Response 1: The sentence in the Abstract section was revised.
Point 2: In the Introduction, the following is inconsistent with the current syndromic definition of essential tremor. I would delete all of it. “In addition to tremor, patients with ET showed additional clinical feature. These include bradykinesia or non-motor symptoms, depression, anxiety, cognitive impairment, fatigue, personality changes, olfactory dysfunction, hearing impairment, temperature extremes, sleep disturbances, and upper airway dysfunction.”
Response 2: According to the Reviewer’s suggestion, the sentence in the Introduction section was erased.
Point 3: Line 54, page 2: “half percent” should be “fifty percent”???
Response 3: According to the Reviewer’s suggestion, we corrected the sentence in the Introduction section.
Point 4: In Table 1: I am not sure what is meant by “worse impairment in the visual-spatial domain”. Is this “visually guided kinetic tremor” or does this refer to cognitive function? Was cognitive impairment linked with the genetic variant.
Response 4: we refer to the lower primary subtest score obtained when intellectual functioning in specific cognitive domains was assessed. According to findings in the literature (#38) where intellectual disability was reported in patients carrying KCNN2 variants, our feeling is that a mild cognitive impairment was linked with the genetic variant identified.
Point 5: All of the affected family members except the proband had ET plus, and the proband had indeterminant tremor due to the short history of tremor (1 year). The proband would eventually be classified as ET plus if his exam remained unchanged for an additional two years. Subject III-3 had antecedent ET or more likely ET plus. Did III-3 have a history of low intelligence? “Essential Tremor” in the title probably should be changed to “Essential Tremor Plus”.
Response 5: we thanks the Reviewer for the helpful comment. According to the Reviewer’s suggestion, we changed the title. As for the subject III-3, we reasoned that because of the presence of Lewy body dementia measures of intellectual functioning would not have been reliable. Therefore, we did not assess this issue.
Point 6: There is no indication of how severe the tremor was in any of the affected family members. Were clinical ratings performed?
Response 6: No. We did not perform clinical ratings to assess the severity of the tremor.
Minor point
Point 1: The paper needs some minor editing to correct English grammar errors.
Response 1: The manuscript was extensively revised to correct English grammar errors.
Reviewer 2 Report
I think the paper looks concise and clearly written. The authors add new conclusive evidence on the new causal genetic variant for familial essential tremor.
I have only one concern: To strengthen the causal association between the KCNN2 variant and the disease status, how about providing the Lod score by linkage analysis?
Author Response
Major points
Point 1: To strengthen the causal association between the KCNN2 variant and the disease status, how about providing the Lod score by linkage analysis?
Response 1: We thanks the Reviewer for her/his comment. We have investigated the KCNN2 genotype in 7 relatives of the nuclear family. Although no recombination between the genotype and the clinical phenotype was observed, we calculated a LOD score of 1,79. As a rule, LOD must be >3.0 to be “significant”. Thus, we supposed that this finding did not add valuable information to readers.
Reviewer 3 Report
Although the article is formally well-written, it appears quite superficial, since it takes fundamental information for granted and, in the clinical approach, authors omitted to consider at least one condition related to tremor, intellectual disability and maternal line transmission. Consequently, the article should not be published before disambiguate some relevant issues.
Major revisions:
1) Introduction: Before considering genetic risks for essential tremor, molecular mechanisms at the basis of the disorder should be explained. In particular, there are not references about ion channels or neurotransmitters role.
2) Material and methods, 2.2. Whole Exome Sequencing, line 101: Had not WES been performed on proband’s parents? If so, which are the reasons to exclude a trio WES?
3) Material and methods, 2.2. Whole Exome Sequencing, line 122-137: it is not clear if variants related to non-neurologic disorders have been excluded from the study.
4) Material and methods, 2.2. Whole Exome Sequencing, line 135-142: these are results and not methods.
5) Results, 3.1. Proband clinical characteristics. Auxological parameters, in particular OFC and height, are lacking.
6) Results, 3.2 Pedigree investigation, line 218-220: “Family history was remarkable for tremor in the mother line [...], suggesting an autosomal-dominant inheritance pattern”. This is a presumption of the authors. Why an X-linked inheritance should be excluded?
In particular, considering that the proband and other members of the family present intellectual disability, with a profile characterized by strengths for verbal skills versus a weakness for visuospatial–constructive skills, weakness in working memory versus strength in processing speed, associated with depression and anxiety, why did authors not exclude FMR1 mutation or premutation [Quintin et al. Dev Psychopathol. 2016 Nov;28(4pt2):1457-1469]?
In addition, the maternal aunt presented a bilateral akinetic-rigid syndrome, and extrapyramidal signs are often signs to suspect Fragile X premutation [Park et al. Brain Behav. 2019 Jul;9(7):e01337].
Generally, essential tremor should be an exclusion diagnosis, having an overlapping presentation with other movement disorders, and being diagnosed initially also in patients with Fragile X premutation [Leehey et al. Arch Neurol. 2003 Jan;60(1):117-21].
FMR1 analysis is thus mandatory before hypothesizing the role of KCCN2 in tremor, especially because this gene has been frequently related to more complex neurologic pictures than tremor.
7) Was brain MRI executed in other members of the family? These data could help identify suggestive pictures and strengthen genotype-phenotype correlations.
Minor editing of English language required
Author Response
We thanks the Reviewer for her/his comments that helped us to ameliorate the presentation of results. We made further changes, and we are confident that the present version of the manuscript is strongly improved.
Major points
Point 1: Introduction: Before considering genetic risks for essential tremor, molecular mechanisms at the basis of the disorder should be explained. In particular, there are not references about ion channels or neurotransmitters role.
Response 1: According to the Reviewer’s suggestion, the Introduction section was revised. Molecular mechanisms at the basis of the disorder and references about ion channels or neurotransmitters role were added (#5, 7, 8, and 9).
Point 2: Material and methods, 2.2. Whole Exome Sequencing, line 101: Had not WES been performed on proband’s parents? If so, which are the reasons to exclude a trio WES?
Response 2: WES was not performed on proband’s parents. According to our internal rules, when a dominant inheritance is assumed, we do not perform a trio WES.
Point 3: Material and methods, 2.2. Whole Exome Sequencing, line 122-137: it is not clear if variants related to non-neurologic disorders have been excluded from the study.
Response 3: The paragraph was rephrased to allow for a better understanding of readers (lines 225-253).
Point 4: Material and methods, 2.2. Whole Exome Sequencing, line 135-142: these are results and not methods.
Response 4: According to the Reviewer’s suggestion, the text is now included in the Results section.
Point 5: Results, 3.1. Proband clinical characteristics. Auxological parameters, in particular OFC and height, are lacking.
Response 5: we thanks the Reviewer for the helpful comment. According to the Reviewer’s suggestion, we added auxological parameters in the Resuls section 3.1 (lines 164-165).
Point 6: Results, 3.2 Pedigree investigation, line 218-220: “Family history was remarkable for tremor in the mother line [...], suggesting an autosomal-dominant inheritance pattern”. This is a presumption of the authors. Why an X-linked inheritance should be excluded?
In particular, considering that the proband and other members of the family present intellectual disability, with a profile characterized by strengths for verbal skills versus a weakness for visuospatial–constructive skills, weakness in working memory versus strength in processing speed, associated with depression and anxiety, why did authors not exclude FMR1 mutation or premutation [Quintin et al. Dev Psychopathol. 2016 Nov;28(4pt2):1457-1469]?
In addition, the maternal aunt presented a bilateral akinetic-rigid syndrome, and extrapyramidal signs are often signs to suspect Fragile X premutation [Park et al. Brain Behav. 2019 Jul;9(7):e01337].
Generally, essential tremor should be an exclusion diagnosis, having an overlapping presentation with other movement disorders, and being diagnosed initially also in patients with Fragile X premutation [Leehey et al. Arch Neurol. 2003 Jan;60(1):117-21].
FMR1 analysis is thus mandatory before hypothesizing the role of KCCN2 in tremor, especially because this gene has been frequently related to more complex neurologic pictures than tremor.
Response 6: we thanks the Reviewer for the helpful comment. FMR1 CGG repeat expansion mutation detection analysis was performed in the proband and his mother. In both of them, genetic analysis revelead the presence of 30(+1) repeats and excluded the presence of FMR1 premutation in the proband and his mother. These findings are presented in the Results section, 3.3 Genetic studies (lines 221-224).
Point 7: Was brain MRI executed in other members of the family? These data could help identify suggestive pictures and strengthen genotype-phenotype correlations.
Response 7: No. We did not perform brain MRI in other members of the family.
Minor point
Point 1: Minor editing of English language required.
Response 1: The manuscript was extensively revised to correct English grammar errors.
Round 2
Reviewer 3 Report
1) Authors should include in the text their choise of excluding a trio WES, as justified in the response to the reviewer.
2) Authors should motivate their assumption about an autosomal dominant transmission, since an X-linked mode of transmission cannot be excluded.
Author Response
We thanks the Reviwer for her/his comments that helped us to ameliorate the presentation of results. We made further changes, and we are confident that the present version of the manuscript is strongly improved.
Major points
Point 1: Authors should include in the text their choise of excluding a trio WES, as justified in the response to the reviewer.
Response 1: According to the Reviewer’s suggestion, the Materials and methods section was revised. The WES strategy used, according to internal rules is now included (lines 107-110).
Point 2: Authors should motivate their assumption about an autosomal dominant transmission, since an X-linked mode of transmission cannot be excluded.
Response 2: According to the Reviewer’s suggestion, the Results section was revised. Reasons that led us to perform WES analys with the assumption of anautosomal dominant transmission are now included (lines 228-231).